# What Helps or Hinders End-of-Life Care in Adult Intensive Care Units in Saudi Arabia? A Mixed-Methods Study Protocol

**DOI:** 10.3390/healthcare12232489

**Published:** 2024-12-09

**Authors:** Nabat Almalki, Breidge Boyle, Peter O’Halloran

**Affiliations:** 1School of Nursing and Midwifery, Queen’s University Belfast, Belfast BT7 1NN, UK; breidge.boyle@qub.ac.uk (B.B.); p.ohalloran@qub.ac.uk (P.O.); 2Nursing Department, Prince Sultan Military College for Health Sciences, Dharan 34313, Saudi Arabia

**Keywords:** end-of-life care, palliative care, intensive care unit, critical care, terminal illness, death and dying, mixed-methods study

## Abstract

**Background**: In the intensive care unit, many patients are dying despite advanced therapeutic technology and optimized treatment. The critical care team is expected to deliver end-of-life care for the dying patient and their family. However, it is challenging to provide adequate support to families due to the ideas and emotions associated with the end of life. This can be influenced by different beliefs and cultures among patients and healthcare professionals. Added to this, research about end-of-life practices in intensive care units in Saudi Arabia is limited. Therefore, this study protocol aims to examine: (1) What end-of-life care is provided by healthcare teams in adult intensive care units in Saudi Arabia? (2) What helps or hinders effective end-of-life care in adult intensive care units in Saudi Arabia? **Method**: This study will use a mixed-methods, sequential, explanatory design consisting of two phases. Phase 1 will include a cross-sectional study design with a convenience sample of approximately 400 healthcare providers who will be invited from three military hospitals in Saud Arabia. the Palliative and End-of-Life Care Index (PEOL Care Index) will be used to assess palliative and end-of-life care education, practice, and perceived competence among the staff. Additionally, a questionnaire on the barriers to optimal end-of-life care and the perceived benefit of potential strategies to improve end-of-life care will be applied to obtain the views of managers. Phase 2 of the study will be a qualitative descriptive design involving semi-structured interviews with healthcare providers, managers, and bereaved family members. **Conclusion**: The study findings will contribute to understanding current practice in relation to palliative and end-of-life care in intensive care units in Saudi Arabia. It will provide valuable insight into barriers to and facilitators of care, which will help to develop strategies and interventions to improve the quality of end-of-life practices in ICUs. In addition, this research will provide significant information regarding family members’ experiences of end-of-life care provided to their relatives.

## 1. Introduction

In intensive care units (ICUs), many patients are dying despite advanced therapeutic technology and optimized treatment [1]. Aldawood et al. [2] reported that 17.6% of patients died in the medical-surgical ICU over a one-year period in a 900-bed tertiary care teaching hospital in Riyadh, Saudi Arabia. This indicates the need to consider end-of-life care alongside interventions to provide continuing intensive care [3,4]. End-of-life care is defined as the physical, spiritual, and psychosocial care provided by health professionals to patients and their families, typically in the last 12 months before death, including after-death support [5].

Ranse et al. [6] described an intensive care environment as a setting where patient treatment and prognosis are uncertain, and the focus is on curative treatments. Nevertheless, the critical care team is expected to deliver end-of-life care for the dying patient and their family [7]. However, it is challenging to provide adequate support to families due to the ideas and emotions associated with the end of life. This can be influenced by different beliefs and cultures among patients and healthcare professionals [8]. Studies reported that cultural norms influence several end-of-life aspects in clinical practice, such as patients’ treatment preferences, planning of death, rituals, family involvement, and provision of care by health professionals. It was found that there is a need for healthcare providers to navigate cultural diversity with sensitivity, ensuring that end-of-life care respects patients’ and families’ unique cultural perspectives while addressing their physical and emotional needs [9,10].

End-of-life care in critical care units has been the focus of many studies in Western countries [3,11]. However, insufficient information is known about end-of-life practices in critical care settings in Middle Eastern countries [12,13]. In our recent systematic review, which searched five databases (Medline, Embase, CINAHL, Psycinfo, and Scopus), we found only nine studies from the Middle East focused on EOLC in ICUs [14]. In many of those countries, end-of-life care is still misunderstood among healthcare professionals, patients, and the community. Beliefs, traditions, and culture have contributed to this confusion, specifically at the end stage of the disease [14,15,16]. Both Arab and Muslim culture make professionals reluctant to withhold any intervention or medication for any patient in the ICU, and Muslim culture and practices are often important to families. Families naturally expect that healthcare is directed to healing and restoring the health of their loved ones, particularly in the highly advanced interventions of the ICU. This, combined with the hope that God will heal the critically ill person at any stage, often creates a social and psychological obstacle to discussing end-of-life care or withdrawing treatments that are no longer effective. Such conversations and decisions may be perceived as actively ending a patient’s life [14].

In Saudi Arabia, with a population of 33.2 million, the life expectancy at birth is 75 years for men and 77 years for women [17]. Saudi Palliative Care National Clinical Guidelines for Oncology were published in 2019 [18]. While the guidelines focus on patients with cancer, they are designed for use by healthcare professionals in any care setting who are involved in supporting people with a palliative life-limiting condition. However, while there has been significant progress in introducing palliative and end-of-life care in the healthcare system in Saudi Arabia, specialist services are not widely available, public knowledge is limited, and there is a persistent focus on curative treatments [19]. Research about EOLC practices in ICUs in Saudi Arabia is limited [2,20]. Thus, this study aims to assess the end-of-life care provided by healthcare professionals in adult intensive care units in Saudi Arabia and to evaluate the facilitators and barriers to providing this care.

### Research Questions

What end-of-life care is provided by healthcare teams in adult intensive care units in Saudi Arabia?

What helps or hinders effective end-of-life care in adult intensive care units in Saudi Arabia?

## 2. Materials and Methods

### 2.1. Study Design

This research will employ a mixed-methods sequential explanatory design, which is used to enhance understanding of quantitative study findings by presenting supporting evidence from the qualitative phase of the research [21,22]. A mixed-methods approach is used in the belief that data gained from the integration of the quantitative and qualitative results provides necessary insights and could enhance practice in ways that go beyond using only one study design [23,24].

Phase 1: Cross-sectional design: A cross-sectional design with ICU healthcare providers and managers will be applied to collect data about the end-of-life care provided in adult intensive care units in Saudi Arabia.

Phase 2: Qualitative descriptive design: A qualitative design involving semi-structured interviews will be applied with intensive care unit (ICU) healthcare professionals, ICU managers, and bereaved family members of the deceased patients. This phase will provide an in-depth explanation and insight into the quantitative data [21]. The main purpose of the qualitative interview is to explore the significant, unexplained, or unexpected results which may be found during the quantitative phase [25]. The results of the quantitative data analysis will be used to direct the qualitative data collection [21]. The results of both the qualitative and quantitative data will be integrated to provide a clear understanding of the current end-of-life care provided in the ICU and its challenges and facilitators. Figure 1 illustrates the mixed-methods research (MMR) procedure that will be used.

### 2.2. Setting

The research will be conducted in adult intensive care units in a sample of military hospitals in Saudi Arabia. Military hospitals contribute to the national healthcare system, participating in medical research, education, and emergency response initiatives, thus enhancing the overall health infrastructure of the Kingdom of Saudi Arabia. In addition, this will be the first EOLC study to be conducted in military hospitals.

Three main military hospitals from different regions of Saudi Arabia will be involved in the study. (1) Prince Sultan Military Medical City (PSMMC) is located in Riyadh city, the capital of Saudi Arabia. It has 76 adult ICU beds of different categories, with approximately 236 critical care nurses. (2) King Fahad Military Medical City is located in Dhahran city in the eastern region. It has 22 adult ICU beds, including a medical, surgical, and cardiac ICU, with around 85 critical care nurses. (3) King Fahad Armed Forces Hospital is located in Jeddah city in the western region. It has 36 adult ICU beds and around 74 critical care nurses. These military hospitals are primarily designed to provide comprehensive medical services to armed forces personnel, their families, healthcare staff, and often the general public. These hospitals are allocated in all regions of Saudi Arabia. They also contribute to the national healthcare system, participating in medical research, education, and emergency response initiatives, thus enhancing the overall health infrastructure of the kingdom [26]. Facilities like the Prince Sultan Military Medical City in Riyadh and the King Fahd Military Medical Complex in Dhahran are among the largest and most advanced, offering specialized medical care.

### 2.3. Study Population and Inclusion Criteria

Healthcare professionals: To be eligible for study recruitment, healthcare professionals must be members of the adult intensive care team and have at least 6 months of experience in critical care in the hospital. With this experience, healthcare professionals are more likely to provide informed insights into EOL practices, challenges, and facilitators. This allows them to become familiar with protocols, multidisciplinary teamwork, and decision-making processes. It also ensures that professionals have engaged with patients and families during emotionally and ethically complex EOL situations, resulting in more relevant and reliable research contributions.

ICU managers: To be eligible for study, the manager must be a nurse or physician director of the adult intensive care unit.

Bereaved family members: To be eligible for the study, they should be a family member of an adult patient who died in the ICU within the last 18 months, above 18 years old, registered in the patient record as next of kin (this may be a friend or other person identified from the records as next of kin), and able to communicate in English or Arabic. The family members will be approached 3–18 months post-bereavement. Including bereaved families within the 3- to 18-month timeframe allows for a variety of perceptions and experiences to be captured. This range ensures that participants who are at different stages of their grief journey can provide diverse insights into their interactions with end-of-life care. Those closer to the 3-month period may offer perspectives influenced by more immediate emotional impacts, while those further along in the 18-month period can provide more reflective insights, adding depth to the study findings. This period of time has been selected to avoid interviews straight after the death of the participants’ family members, as the discussion of the death of the loved ones is an emotive topic. Previous research suggests that there is a low likelihood of distress among family members participating in end-of-life care survey research when allowing at least six weeks to reduce the potential of any distress in receiving a survey so close to the patient’s death [27]. The approach will be within the time frame that will allow them to remember the experiences but will reduce the risk of distress [28].

### 2.4. Sampling and Sample Size

Phase 1: Cross-sectional design

The anticipated total number of adult ICU healthcare providers in the three hospitals selected is around 400. Therefore, all individuals who meet the inclusion criteria will be eligible to be recruited for this study to allow an adequate sample, as larger sample sizes give more reliable results. Likewise, all ICU nurses and physician managers who meet the inclusion criteria will be eligible for the study. A convenience sampling technique will be used to recruit the participants in the quantitative phase of the study [29]. According to Fincham [30], to ensure a representative sample for most research, response rates of around 60% should be the goal of researchers. Based on the range of responses in previous studies of EOLC, it is estimated that a 60% response rate from healthcare providers should be achieved [31,32].

Phase 2: Qualitative design

As noted above, the primary purpose of semi-structured interviews is to allow the researcher to explore and clarify significant aspects related to the research questions through active conversation with participants. Thus, the sample is intended to provide sufficient numbers to achieve understanding in combination with the quantitative data already collected and analyzed [33].

This phase will use purposive sampling to recruit approximately 15 healthcare professionals including nurses, physicians, nurse managers, and physician managers. In addition, six to nine family members will be recruited.

### 2.5. Recruitment and Data Collection

Phase 1: Unit supervisors of ICUs will be approached through email and will be involved as gatekeepers in the process of recruiting the ICU professionals. If the healthcare providers/managers meet the inclusion criteria, the study information, online survey, and researcher contact details will be provided to participants through gatekeepers. Explanations for any unclear concept in the questionnaire will be provided by the researcher. We will ensure participants are fully informed about the study by including a participant information sheet as part of the introduction to the survey. Responding to the survey will then be taken as consent to take part in the study. If participants have not returned the questionnaire within two weeks, a reminder email will be sent through the unit managers to maximize the response rate. If the response remains low, another reminder will be sent after two weeks. The most effective strategy to have a high response rate from healthcare professionals is the use of repeat mailings [34,35].

The survey will be distributed using Microsoft Forms to collect data through an online link. Microsoft Forms, accessed through Queen’s University Belfast, allows researchers to create online surveys easily and quickly. At the same time, it is a secure software that ensures confidentiality, integrity, easy availability, and privacy of information processing according to the international standards stated in the company’s privacy statement [36].

Phase 2-A: Bereaved family members

The ICU manager will assist in the process of identifying the bereaved family members of patients who died in the last 3–18 months from the death records. The ICU manager will act as a gatekeeper who communicates with bereaved family members through telephone contact to introduce the researcher and to ensure the family members’ eligibility and interest in participation. Then, the researcher will contact the family members who are willing to participate by telephone to provide them with interview information, consent, and research contact details. After one day, the researcher then will approach the participants to confirm if they need any further clarification and whether they are willing to participate. The interview will be arranged once the written consent is secured from participants. The interview will be conducted through a voice call and digitally recorded. The interview will take approximately 20–30 min.

Phase 2-B: Healthcare professionals and ICU managers

All participants who take part in the survey will be invited to participate in an additional interview. Staff who are interested and willing to be contacted for the interview will be asked to give their contact information in another recruitment link provided with the survey link through gatekeepers. The researcher then will contact the participants to provide them with a consent form and study information sheet explaining the purpose of the interview. The virtual interview will be arranged once the consent is obtained. Interviews will be conducted in Microsoft Teams and take approximately 20–30 min.

### 2.6. Instruments and Data Sources

Phase 1: Healthcare professionals’ questionnaire

Palliative and end-of-life care practices will be assessed using the Palliative and End-of-Life Care Index (PEOL Care Index). The PEOL Care Index is designed to assess palliative and end-of-life care education, practice, and perceived competence among intensive care unit (ICU) nurses [31]. Minor modifications of terms used in the questions will be made to include physicians, such as adding the discipline category of nursing or medicine and modifying the term “nursing intervention” in items (5, 7, 20) to the general term “appropriate/therapeutic intervention”. The PEOL Care Index was developed and judged in a previous study for content validity by international panellists and then pretested in a pilot study, where data were collected at two time points using self-administered questionnaires, followed by cognitive interviews [31]. Test–retest reliability was examined by Eltaybani, Igarashi [31] using intraclass correlation (ICC), standard error of measurement (SEM), and repeatability coefficient (RC). This questionnaire was tested in Egypt which is an Arabic country and one of the Middle Eastern countries. The questionnaire consists of two parts. The first part presents demographic data, including age, gender, years of nursing experience, years of ICU experience, qualification, work-related shift data, previous caring of dying patients, the stress of dealing with death and dying, and in-service training or education on PEOL care. The second part adopts the definition and domains stated by the National Consensus Project for Quality Palliative Care [37]. The eight domains are (1) structure and process of care; (2) physical aspect of care; (3) psychological and psychiatric aspects of care; (4) social aspect of care; (5) spiritual, religious, and existential aspects of care; (6) cultural aspect of care; (7) care of the patient nearing the end of life; and (8) ethical and legal aspects of care. It consists of 25 items with three Likert scales from 0 to 5 to assess education, practice, and perceived competence of each item (where for education, 0 = no education and 5 = extensive education; for practice, 0 = never and 5 = always; and for competence, 0 = no competence and 5 = almost perfect) [31]. The time needed to complete this questionnaire is approximately 15 min.

Phase 1: ICU managers’ questionnaire

The barriers to providing optimal end-of-life care in the intensive care unit and strategies to improve the care will be assessed using a questionnaire developed by Nelson et al. [38]. The questionnaire will be applied to obtain the views and experiences of ICU managers about the barriers to optimal end-of-life care and the perceived benefit of potential strategies to improve this care. This questionnaire was developed in a previous study by Nelson et al. [38] by 43 international and interdisciplinary clinicians, investigators, and educators with clinical and research experience in end-of-life care. It was then judged for content validity by pretesting, clinical sensibility assessment, inter-rater reliability testing, and then pilot testing. The questionnaire has three sections. The first part consists of demographic data, including discipline, gender, age, speciality/subspeciality of physicians, duration of critical care practice, type of ICU, size of ICU, and hospital name. The second part consists of three domains to classify possible barriers to the optimal care of patients dying in the ICU: patient/family factors (7 items), clinician factors (14 items rating the roles of both physicians and nurses), and institutional/ICU factors (11 items). The respondents are asked to what extent these factors were barriers (5-point scale ranging from 1, a huge barrier, to 5, not a barrier at all). The third section involves 14 strategies to improve end-of-life care in the intensive care unit.

Phase 2: Semi-structured interviews will be applied to allow the researcher to explore and clarify significant aspects related to the research questions through active conversation with participants.

Initially, interviews will be conducted with bereaved family members and then with ICU staff. Most of the interview questions will be formulated after the initial analysis of quantitative data to provide clarification of unexplained or salient findings. However, the interview questions for healthcare providers will be formulated from both the quantitative data analysis and the information learned from bereaved family interviews.

### 2.7. Data Analysis

Phase 1 will apply a descriptive statistics analysis using IBM SPSS Statistics software Version 29.0.0.0 (241) [39]. The mean and standard deviation will be reported for normally distributed data. Alternatively, the median and interquartile range will be determined for skewed data. For categorical variables, the frequency and percentage of the response will be presented. As the online survey is designed only to allow submission when all items have been completed, we do not anticipate having missing data. By examining the distribution of frequencies and scores of responses, researchers will determine patterns and trends that indicate significant or unexpected insights. This will allow the researchers to prioritize areas for further investigation and draw meaningful conclusions.

Phase 2 will used thematic analysis to analyze the transcripts. Individuals’ interview records will be transcribed by the researcher; the Arabic interviews with the bereaved family members will be translated into English by the researcher. The translated transcripts will then be checked independently by another bilingual official translator to confirm the quality of the transcription’s translation. A comprehensive reading of the transcripts will be applied to develop accurate themes and coding that reflect the data set of the interview texts. This will be achieved through applying the six phases of analysis developed by Braun and Clarke [40,41]: familiarisation with the data, coding, identifying themes, reviewing themes, classifying and defining the themes, and reporting. Interviews will be analyzed by the three members of the research team; subsequently, findings will be discussed and verified by the team at every stage to assess the accuracy of the interpretation, improve reliability, and ensure a rigorous qualitative phase for the research. Data will be coded and managed by the NVivo qualitative data analysis software (https://lumivero.com/) [42].

### 2.8. Integration

Integration is the stage of mixed-methods research where the collected quantitative and qualitative data are integrated, linked, or mixed. This process is an essential step in a mixed-methods design [43]. Integration in this research will be at the design, methods, and interpretation stages. The integration at the design level will occur by using the explanatory sequential mixed-methods design [21]. At the methods level, integration will be through building on quantitative findings to inform the data collection approach of the qualitative phase. The interpretation level of integration will be achieved through a narrative approach to synthesize the study findings from the two sets of data [23]. This process of integration is illustrated in Figure 2.

### 2.9. Rigour

Plano Clark [44] recommends that researchers should consider rigour when conducting mixed-methods research. This study will apply three quality standards to ensure rigour: standards for quantitative methods, qualitative methods, and mixed methods. There are various criteria to ensure the rigour of each of these research methods. The quantitative research criteria are validity, reliability, replicability, and generalisability, while the qualitative research criteria are credibility, transferability, dependability, and confirmability [45,46]. In mixed-methods research, the criteria of quantitative and qualitative research will be applied separately [23]. In addition, this study will use the guidelines of Good Reporting of a Mixed-Methods Study (GRAMMS) to enhance the transparency of the study processes and guide the quality of reporting (see Table 1) [47,48].

## 3. Discussion

The study findings will contribute to understanding current practice in relation to palliative and end-of-life care in intensive care units in Saudi Arabia. It will provide valuable insight into barriers and facilitators of care. This will also provide valuable information and insight for healthcare teams wishing to develop strategies and interventions to improve the quality of EOLC practices in ICUs. In addition, this research will provide significant information regarding family members’ experiences of EOLC provided to their relatives in ICUs. This finding will lead to a better understanding of the needs of patients’ families during the end-of-life phase. However, a key limitation is the challenges of recruiting bereaved family members due to the emotional sensitivity of the topic, which may deter participation and introduce selection bias [49].

## 4. Conclusions

This mixed-methods protocol aims to investigate end-of-life care practices in Saudi Arabia to identify key barriers and facilitators influencing care in intensive care units. Using a sequential explanatory design, the research will integrate quantitative surveys of ICU staff and managers with qualitative interviews involving bereaved families and ICU staff. The study aims to uncover culturally relevant insights into current end-of-life care practices, emphasizing the perspectives of families and healthcare providers. The findings will inform the development of national guidelines that align with Saudi cultural and healthcare contexts. It will also inform the educational and training programs for healthcare staff based on these guidelines to improve the quality of end-of-life care.

## 5. Ethical Considerations

A consent form is required from all participants involved in the study. During the interview, discussion of end-of-life care may cause distress, as it is an emotive topic. If any participant experiences distress during the interview, the researcher will put the interview on hold until given verbal consent to return to the interview by the participant. Alternatively, the interview will be terminated if the researcher feels it would be distressing for a participant to re-engage with the interview, and the participant will be advised to contact their primary healthcare services for additional support as required [50,51,52]. In addition, participants will be free to withdraw from the study at any time without any disadvantage and without any obligation to give a reason, and all responses and information relating to a participant who has withdrawn will be destroyed and removed from all study documents.

Participant identification data and responses will be anonymized. All data will be stored securely on a computer protected with a password that is known only by the researcher. When the interview has been transcribed, the recorded tape will be destroyed.

## Figures and Tables

**Figure 1 healthcare-12-02489-f001:**
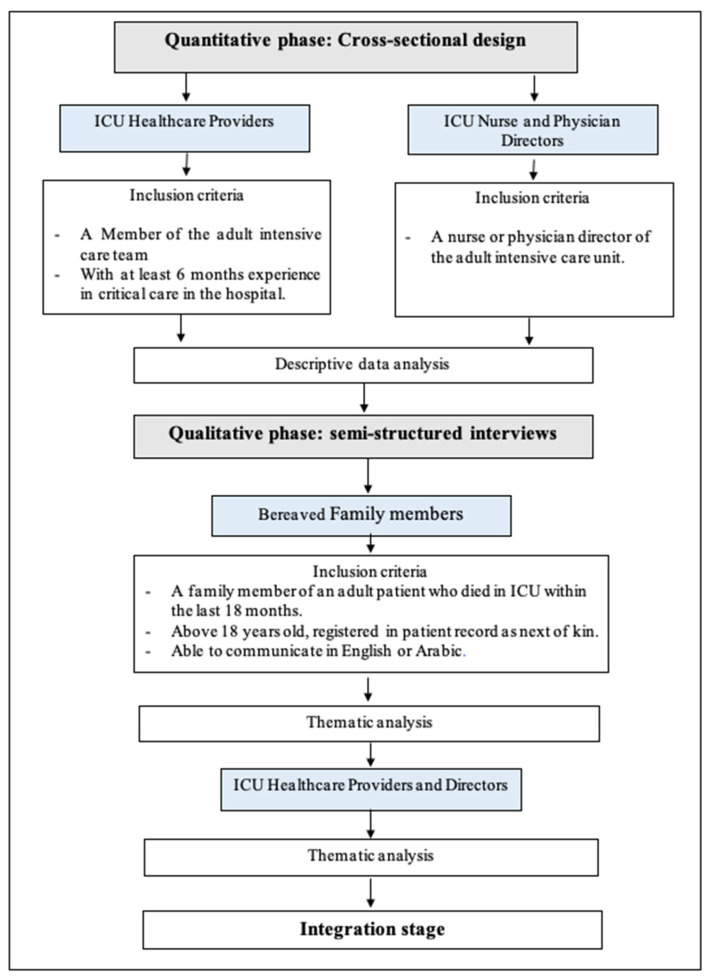
Diagram of study procedures.

**Figure 2 healthcare-12-02489-f002:**
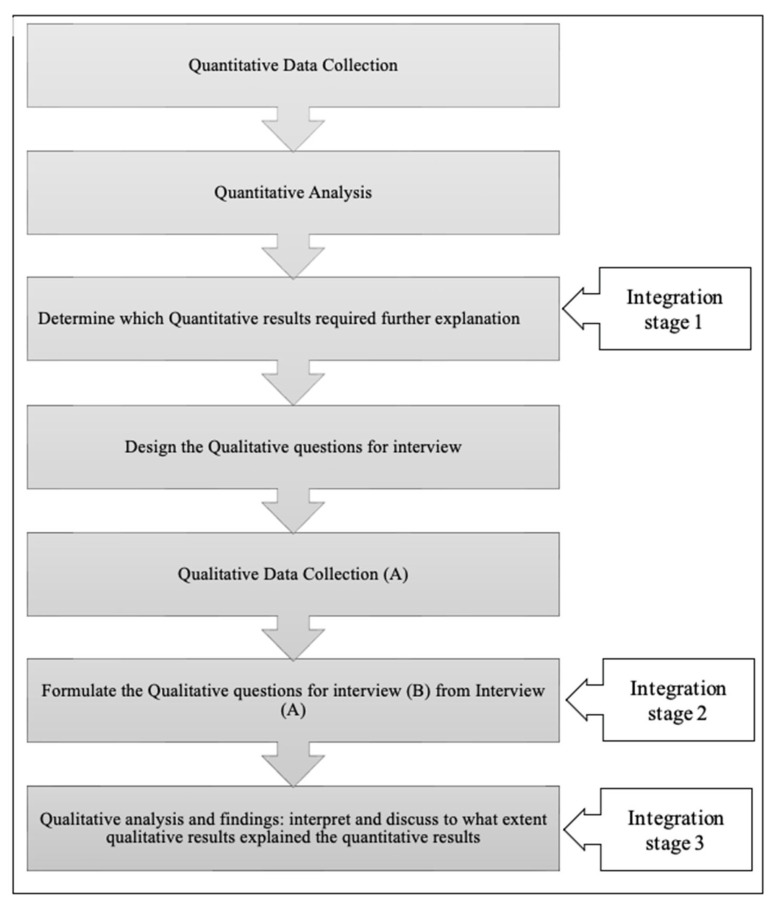
Flowchart of the implementation and integration of explanatory sequential mixed-methods design.

**Table 1 healthcare-12-02489-t001:** The GRAMMS guidelines as described by O’Cathain et al. [47].

(1) Describe the justification for using a mixed-methods approach to the research question.
(2) Describe the design in terms of the purpose, priority, and sequence of methods.
(3) Describe each method in terms of sampling, data collection, and analysis.
(4) Describe where integration has occurred, how it has occurred, and who has participated in it.
(5) Describe any limitation of one method associated with the presence of the other method.
(6) Describe any insights gained from mixing or integrating methods.

## Data Availability

This protocol does not contain data. All data will be available in the manuscript.

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
