# Peer review of "What Helps or Hinders End-of-Life Care in Adult Intensive Care Units in Saudi Arabia? A Mixed-Methods Study Protocol"

_healthcare, 2024, doi:10.3390/healthcare12232489_

Round 1
Reviewer 1 Report
Comments and Suggestions for Authors
Well done and effort on formulating and conducting this study protocol. This topic "End-of-Life Care in Adult Intensive Care Units" is indeed and still there is a gap in research in particular SA. Some feedback that might improve the manuscript and to be ready for publication based on the journal criteria:
I suggest to rewrite the title or modify it as you published a systematic literature review and the title is similar "Almalki, N., Boyle, B., & O’Halloran, P. (2024). What helps or hinders effective end-of-life care in adult intensive care units in Middle Eastern countries? A systematic review. BMC Palliative Care, 23(1), 87."
1- Abstract: make sure that you are listing the words complete before using the abbreviations such as ICU and be consistent in the whole paper. In the method could you add the expected calculated sample size (healthcare providers, managers, and family members). Also, if you could shorten the abstract to 350 as it is exceeds more than 400.
2- Introduction: watch some type and grammatical mistakes such as in line 39 repeated in's. Be consistent in intext citation references Chicago vs APA format and follow the journal requirement. In line 51-53, it is better to add relevant studies that shows how cultures and emotions might influences both families and healthcare givers. in line 55-60, it is interesting to find out that believes, traditions and cultures as well contribute to misunderstanding the EOL however you have to support this with studies and elaborate more with study results as an example. In line 62, if there is recent reference to life expectancy in SA added it ( as the supported one 2016)
3- Material and method: is the is a purpose from targeting only military hospitals? give a strong justification as you mention Culture and believes in in the introduction or is it support the EOL protocol and practices.
Line 120: in the inclusion, give rational why 6 month experiences in critical unit
sampling and sample size: need to clarify how did you calculate the sample size.
Instrument: well done and comprehensive
Data analysis: add how you going to keep the data confidential and how you going to handle missing data
line 333-342, remove as it is journal instructions
references: I noticed that many references were considering old, kindly update the references within 5 y
Author Response
We thank reviewer 1 for recognising the relevance of the protocol, and for all the comments and suggestions, which have helped us to strengthen the paper.
Please see the attachment point-by-point response.

Reviewer 2 Report
Comments and Suggestions for Authors
Dear Authors,
I commend you on the topic addressed in your research. The protocol is clear; however, I have a few suggestions to improve the quality of the manuscript:
- Please review the references. Ensure each includes the DOI or a web link to the publication.
- Of the 45 references cited, only 18 (or 40%) have been published within the last five years. It is recommended that you update references in the manuscript to better reflect the current state of the art on the subject.
- In the introduction, regarding the following statement: “However, insufficient information is known about end-of-life practices in critical care settings in Middle Eastern countries [1, 10, 11],” on which database was this search conducted? What keywords were used?
- In the qualitative stage, the authors mention that an inclusion criterion is family members of an adult patient who died within the past 18 months. Did you consider that this period (18 months) might be too long, potentially introducing recall bias?
- In Phase 1, I believe it would be appropriate for the researchers to obtain written consent from participants.
Author Response
We thank reviewer 2 for recognising the relevance of the protocol, and for all the comments and suggestions, which have helped us to strengthen the paper.
Please see the attachment point-by-point response.

Reviewer 3 Report
Comments and Suggestions for Authors
This study protocol addresses an important and timely topic: end-of-life care (EOLC) in adult intensive care units (ICUs) in Saudi Arabia. As a young nation with a median age of 30 years, Saudi Arabia must proactively prepare for the challenges of an aging population, ensuring high-quality, culturally and contextually appropriate palliative care services. The authors’ focus on ICU settings, where curative treatments often dominate, is particularly relevant given the growing recognition of the need for EOLC integration.
The methodology, based on a mixed-methods sequential explanatory design, is sound and aligns well with the study’s objectives. While cross-sectional designs have inherent limitations, this foundational study is an appropriate and valuable starting point in an under-investigated field. The involvement of diverse stakeholders—healthcare professionals, managers, and bereaved family members—adds depth and breadth to the research, providing a comprehensive understanding of EOLC in ICUs.
Suggestions for Improvement:
1. Clarify Inclusion of “Family Caregivers”:
The protocol mentions “family members,” but it is important to note that family caregivers may not always be biological family members; they may also be neighbors or friends, particularly in bereavement contexts. While this is common in Western settings, cultural nuances in Saudi Arabia may differ. Expanding the study protocol to acknowledge and potentially include these individuals would enrich the findings.
2. Cultural Context:
The study acknowledges the influence of culture on EOLC but could benefit from explicitly addressing how Saudi Arabia’s unique cultural and religious context may shape family and healthcare professional interactions. For instance, understanding how Islamic beliefs about death and dying influence EOLC practices would add depth to the analysis.
3. Future Directions:
While the study aims to identify barriers and facilitators, recommendations for sustainable, culturally aligned interventions should also be emphasized as a goal. This could include exploring how findings could inform national guidelines or training programs for healthcare professionals.
4. Limitations and Challenges:
The authors might briefly discuss the challenges inherent in recruiting bereaved family members, as this can be an emotionally sensitive topic. Addressing potential biases or recruitment hurdles upfront would strengthen the protocol’s transparency.
This protocol lays critical groundwork for advancing EOLC in Saudi Arabia, providing a foundation for future longitudinal or interventional studies. Its mixed-methods approach and inclusion of diverse stakeholders are strong points, and with minor adjustments, the study has the potential to significantly contribute to improving palliative care in resource-limited, culturally distinct settings.
Author Response
We thank reviewer 3 for recognising the relevance of the protocol, and for all the comments and suggestions, which have helped us to strengthen the paper.
Please see the attachment point-by-point response.
